# Reference Trustable Decoding: A Training-Free Augmentation Paradigm for Large Language Models

**Luohe Shi**[1,†], **Yao Yao**[2], **Zuchao Li**[1,†,*] **Lefei Zhang**[1], **and Hai Zhao**[2]

[1]National Engineering Research Center for Multimedia Software,
School of Computer Science, Wuhan University, Wuhan, 430072, P. R. China
[2]Department of Computer Science and Engineering, Shanghai Jiao Tong University
`shiluohe@whu.edu.cn, yaoyao27@sjtu.edu.cn, zcli-charlie@whu.edu.cn,`
`zhaohai@cs.sjtu.edu.cn`

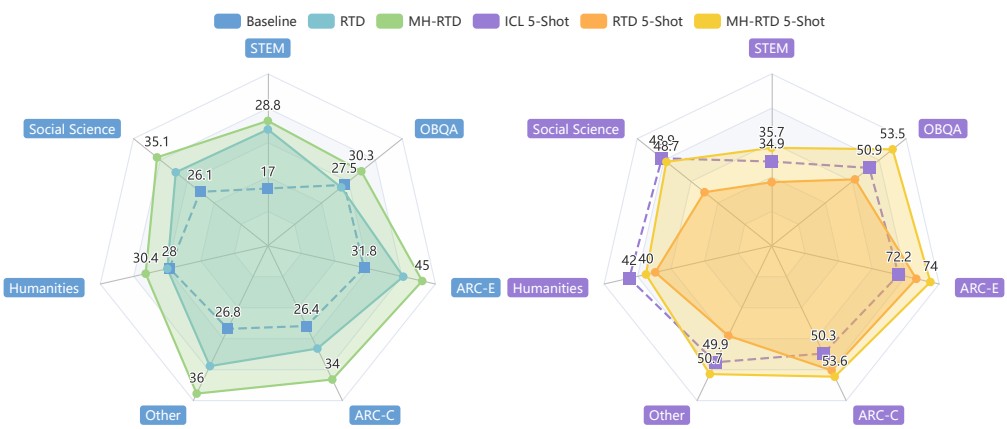

Figure 1: Performance comparison between default LLM and reference trustable decoding in reasoning tests.

## Abstract

Large language models (LLMs) have rapidly advanced and demonstrated impressive capabilities. In-Context Learning (ICL) and Parameter-Efficient Fine-Tuning (PEFT) are currently two mainstream methods for augmenting LLMs to downstream tasks. ICL typically constructs a few-shot learning scenario, either manually or by setting up a Retrieval-Augmented Generation (RAG) system, helping models quickly grasp domain knowledge or question-answering patterns without changing model parameters. However, this approach involves trade-offs, such as slower inference speed and increased space occupancy. PEFT assists the model in adapting to tasks through minimal parameter modifications, but the training process still demands high hardware requirements, even with a small number of parameters involved. To address these challenges, we propose Reference Trustable Decoding (RTD), a paradigm that allows models to quickly adapt to new tasks without fine-tuning, maintaining low inference costs. RTD constructs a reference datastore from the provided training examples and optimizes the LLM's final vocabulary distribution by flexibly selecting suitable references based on the input, resulting in more trustable responses and enabling the model to adapt to downstream tasks at a low cost. Experimental evaluations on various LLMs using different benchmarks demonstrate that RTD establishes a new paradigm for augmenting models

---

\* Corresponding author. † Equal contribution.

38th Conference on Neural Information Processing Systems (NeurIPS 2024).

to downstream tasks. Furthermore, our method exhibits strong orthogonality with traditional methods, allowing for concurrent usage. Our code can be found at https://github.com/ShiLuohe/ReferenceTrustableDecoding

# 1 Introduction

In the rapidly advancing field of artificial intelligence, Large Language Models (LLMs) have demonstrated substantial progress. With their extensive parameter size, LLMs have acquired emergent abilities [41] and been able to tackle diverse and challenging tasks in fields like education [22] and medicine [38]. Despite their immense potential, Large Language Models that have just completed pre-training often struggle to effectively adapt to downstream tasks. Moreover, the process of adapting the model is typically costly and requires careful execution by experienced individuals. Otherwise, it could lead to the model generating hallucination [50; 28] at best, or at worst, result in a loss of its language capabilities.

In-Context Learning (ICL), as a category of methods that do not require parameter adjustments, is one of the mainstream methods for adapting models to downstream tasks. ICL embeds domain knowledge, question-answering patterns, etc., into prompts through few-shot learning [6], prompt engineering [51], and Retrieval-Augmented Generation (RAG) [26] methods, leveraging the learning ability of the model itself to provide better answers. As pointed out in Figure 2, ICL focuses on the prompt stage. However, ICL significantly increases the length of the input, consequently increases the space occupied by the KV-Cache required for inference. Further, according to the Roofline model [46], this part of the KV-Cache cannot be parallelized through batch processing, making memory I/O throughput a system bottleneck, wasting hardware computing power, and increasing token generation time during the entire inference stage.

Fine-tuning is also used to adapt models to downstream tasks. By fine-tuning the pre-trained model based on domain tasks, the model can quickly acquire capabilities within the domain. However, traditional full-parameter fine-tuning often requires a large amount of resources (empirically 8-15 times that of inference), making Parameter-Efficient Fine-Tuning (PEFT) a more popular method. By freezing most parameters and only modifying a few, methods such as Adapters, P-tuning [27], LoRA [16] and others [48; 36; 44] have become mainstream methods for quickly adapting models to downstream tasks. However, fine-tuning methods introduce several hyperparameters, which require high experience from the fine-tuners and the effects are unpredictable. Furthermore, due to the need for backpropagation, the computation graph must be saved, meaning that even if only a few parameters need to be updated, there will be a large amount of additional computation and space requirements (several times that of inference), raising the threshold for methods based on fine-tuning.

To address these challenges, we introduce Reference Trustable Decoding (RTD), a novel framework designed to fit LLMs for downstream tasks. Distinct from a conventional `LM_Head` module, RTD strategically retrieves relevant references from a pre-constructed datastore, guided by the final hidden states of the language model. This approach not only enhances the final output distribution by recalculating it with the similarity score of the retrieved references but also allows for the seamless integration of new knowledge or constraints into the response generation process without increasing the input length or using gradient descent.

RTD, distinctively training-free, emphasizes compact input lengths to expedite inference. RTD's effectiveness was rigorously tested using varied benchmarks focused on different tasks and a Wikipedia-based knowledge injection scenario. On these benchmarks, RTD achieved results comparable to traditional methods like PEFT and ICL, providing significant improvement. Additionally, we combined RTD with traditional methods, further enhancing the model's capabilities and demonstrating the good orthogonality of RTD with other approaches.

Our contribution includes:

- We propose a new paradigm, called RTD, for fitting LLMs for downstream tasks. RTD is a training-free method that focused on the decoding stage of large language models (LLMs), as a alternation of `LM_Head`. It helps LLMs to adapt to different tasks with different demands and provide trustable response.

- RTD has achieved performance comparable to, or even better than, ICL and PEFT across different benches, while maintaining the desirable properties of training free and not intro-

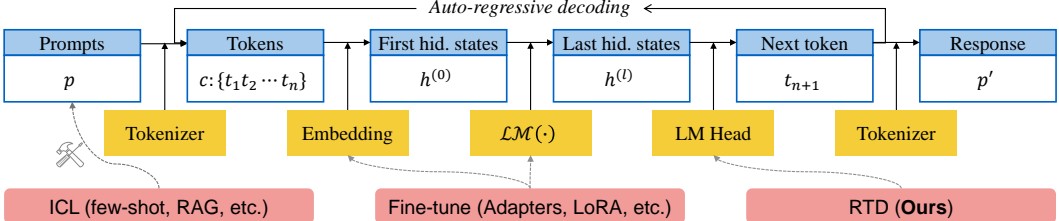

Figure 2: The pipeline of LLM inference and the focus of different methods: ICL focuses on the prompt stage, emphasizing the optimization of the model's input. Fine-tuning methods optimize the model itself by adjusting its parameters. In contrast, our proposed RTD method targets the decoding stage of the language model. By constructing a reference datastore, RTD optimizes the final output distribution without requiring additional training.

ducing additional input lengths. This demonstrates the potential of RTD as a new paradigm for LLMs to adapt to downstream tasks. Furthermore, RTD can be seamlessly integrated with other existing methods, such as in-context learning (ICL) and fine-tuning. The combination of RTD, ICL, and fine-tuning has the potential to achieve even higher performance.

## 2 Background and Related Work

In the field of NLPs, Transformer models have gained influence rapidly after it get original proposed in 2017. As larger scaled model been introduced, especially giant ones like GPT3 which has 175 Billion of parameters [9], the training process is getting more and more expensive, hence to fit the LLMs for downstream tasks.

### 2.1 Fine-tuning

**Full Parameter Fine-tuning**   Full parameter fine-tuning refers to fully optimizing all the parameters of the model during the fine-tuning process. Full parameter fine-tuning has the advantage of allowing the model to adapt more closely to the specific task at hand, as well as injecting more information into the model. However, it also has the disadvantage of being the most computationally expensive and time-consuming, as it requires to manipulate all parameters of the model, with the modern optimizer like Adam [19], 8 to 15 times of more extra GPU memory is demanded comparing to inference empirically, resulting a must of multi-GPU server or even cross sever training.

**Parameter Efficient Fine-tuning**   Parameter Efficient Fine-tune (PEFT), for example, LoRA [16] and P-tuning [27], is introduced to make fine-tune more reachable. By freezing most of the model parameters and only let a small amount of them accumulate gradient, the GPU memory and computation resource can be cut down by a large margin [14].

However, as fine-tuning introduce many tricky hyper-parameters like learning rate, the process is heavily task related and empirical, even experienced fine-tuner need some trials and error when tuning them. Moreover, even if the number of parameters trained is not large, processes such as backpropagation still need to be carried out. The computation graph generated on long sequences will also occupy a large amount of memory, making the threshold for computing power and memory still high, which any method that relies on gradient descent is difficult to avoid.

### 2.2 In-Context Learning

**Few-Shot Learning**   Few-shot Learning is proved to be a great way for LLMs to gain capability. By appending the true task that LLMs are expecting to response after a couple of existing correct examples, LLMs can gain its reasoning ability [6; 31].

**Retrieval Augmented Generation**   Retrieval Augmented Generation (RAG)  [24] is an AI framework for retrieving facts from an external knowledge source to LLMs, which helps LLMs correct its hallucination and use latest fact [35]. RAG is to cut external knowledge source into multiple chunks,

then embed and store them in a database, then retrieve them at the process of generation to let LLMs get the knowledge in it. This technique allowed LLM to use extra information while maintaining their parameters untouched. RAG have been used on multiple fields, like coding [26] and question answering [29]. And it can be combined with few-shot [18].

The main drawback of ICL methods lies in their growth of the input sequence. Under the quadratic complexity of the Transformer architecture, this implies a longer KV-Cache, which not only increases the latency during the pre-fill stage but also adds delay each time a token is generated [45]. Moreover, unlike model parameters, each instance needs to save its dedicated portion of KV-Cache, leading to memory I/O bottlenecks and computational power waste. Finally, on some smaller models, the irrelevant information that ICL might contain can confuse the model, resulting in performance loss.

## 3  Reference Trustable Decoding

In this section, we begin by presenting the fundamental formulas and concepts to elucidate the workings of Reference Trustable Decoding, followed by an exploration of the multi-head Reference Trustable Decoding method.

### 3.1  Preliminary

Given an input sentence $c = \{t_1, t_2, ..., t_n\}$, where $t_i$ represents the i-th token and $n$ denotes the sentence length, the last token's output of the last Transformer block in the language model can be represented as:

$$h^{(l)} = \mathcal{LM}(c) \tag{1}$$

In this equation, $h^{(l)} \in \mathbb{R}^{d_m}$ is the output of the last token from the final, or the $l$-th, Transformer block of the language model, where $d_m$ denotes the hidden size of the model.

Traditionally, a standard decoder-only architecture Transformer usually employs `LM_Head`, which is, a fully connected layer, usually includes a learnable weight matrix $W$ and no bias, followed by a softmax function $\mathbf{Softmax}(\cdot)$ to predict the output probability distribution $\mathbf{p}$ of the next token from the last hidden states:

$$\mathbf{p} = \texttt{LM\_Head}(h^{(l)}) = \mathbf{Softmax}(W \cdot h^{(l)}) \tag{2}$$

where $v$ is the vocabulary size and $W \in \mathbb{R}^{v \times d_m}$.

However, traditional next token prediction does not support incorporating external information and therefore, we introduce reference trustable decoding where we build a bypass around the `LM_Head`, showcased in Figure 3, as the entrance of additional knowledge or guidance.

### 3.2  Reference Trustable Decoding

#### 3.2.1  Generation of Reference Datastore

In reference trustable decoding, we first build the reference datastore $\mathcal{L}$, which stores key-value pairs $(k, v) \in (\mathcal{K}, \mathcal{V})$. Here, the key $k = \mathcal{LM}(c)$ represents the last hidden states of the token generated by the LMs from the context $c$, and the value $v$ is the corresponding label $y$. Mathematically, we have:

$$\mathcal{L} = \{(k, v)|(k, v) \in (\mathcal{K}, \mathcal{V})\} = \{(\mathcal{LM}(c), y) \mid (c, y) \in \mathcal{D}\} \tag{3}$$

where $\mathcal{D} = (\mathcal{C}, \mathcal{Y})$ is the task dataset with input context set $\mathcal{C}$ and label set $\mathcal{Y}$, and $|\mathcal{Y}|$ refers the number of possible labels. This process is depicted in Figure 3. It's obvious that **the computational requirement is same as performing a forward pass to every content in the task dataset**, which aligned with the minimal requirement of the inference stage, denotes the superiority of RTD as a gradient-free method.

#### 3.2.2  Decoding Stage

At each decoding round, given the input context $c$, we first compute $h^{(l)} = \mathcal{LM}(c)$, which is the input to RTD and `LM_Head`. Then we use a three stage approach to get the RTD output, **Fetch**, **Normalization**, and **Aggregation**, depicted in Figure 4.

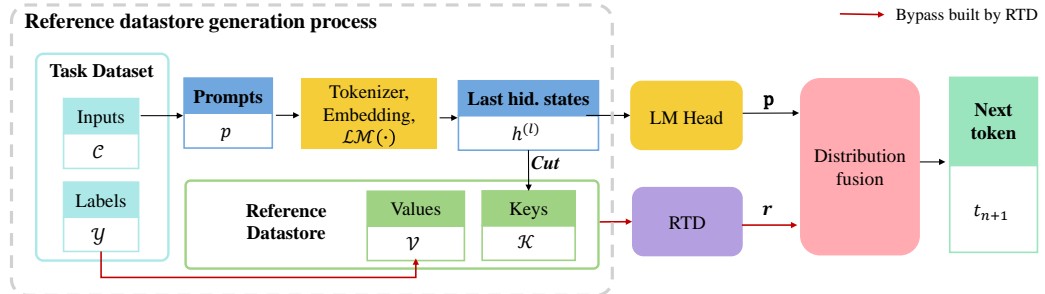

Figure 3: Overview of the reference datastore generation and reference trustable decoding process.

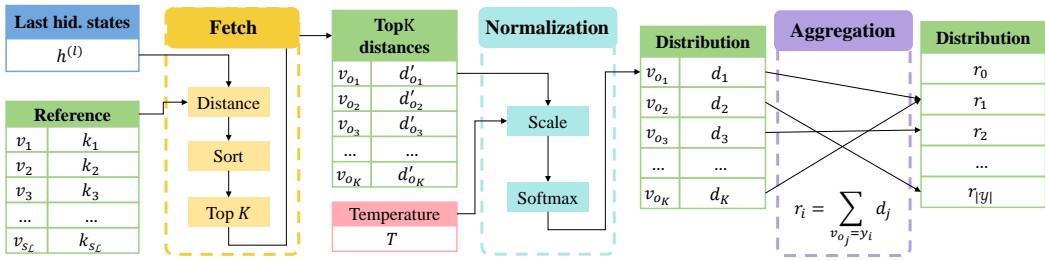

Figure 4: Three stages of reference trustable decoding.

**Fetch**   First, we calculate the distance $d'$ between $h^{(l)}$ and all the $k$ in the reference datastore $\mathcal{L}$. Otherwise stated, we use the Euclidean distance $d'_i = ||h^{(l)} - k_i||_2$. We then select the top $K$ instances from $\mathcal{L}$ which have the smallest distance, and for the $j$-th ($1 \leq j \leq K$) closest $(k_i, v_i)$, we define $o_j = i$. Then we create a set $L_h$, storing the top $K$ distances and values:

$$L_h = \{(d'_{o_j}, v_{o_j})\} = \{(||h^{(l)} - k_{o_j}||_2, v_{o_j})\}, \quad o_j = i \text{ for } j-\text{th closest } (k_i, v_i) \tag{4}$$

**Normalization**   We first scale the $d'$ we got from the previous stage by temperature $T$, as $d''_j = d'_{o_j}/T$. The scale operation is introduced to prevent overflow in the following Softmax operation. We take the Softmax of $-d''$ as $d$, guaranteed $d$ as a valid possibility distribution.

$$d = \textbf{Softmax}(-d''), \quad d_j = \frac{\exp\{-d''_j\}}{\sum_{\iota=1}^{K} \exp\{-d''_\iota\}} = \frac{\exp\{-d'_{o_j}/T\}}{\sum_{\iota=1}^{K} \exp\{-d'_{o_\iota}/T\}} \tag{5}$$

**Aggregation**   We calculate the final reference possibility distribution $\mathbf{r} = [r_1, r_2, ..., r_{|\mathcal{Y}|}] \in \mathbb{R}^{|\mathcal{Y}|}$ by aggregating all $d_j$ that satisfies $v_{o_j} = y_i$, where $y_i \in \mathcal{Y}$.

$$r_i = \sum_{v_{o_j} = y_i} d_j \tag{6}$$

We denote $\mathcal{R}(\cdot, \mathcal{L}) : \mathbb{R}^{d_m} \to \mathbb{R}^{|\mathcal{Y}|}$ as the function represents all three stages of querying the datastore $\mathcal{L}$ and building the corresponding reference possibility distribution $\mathbf{r}$. Therefore, we have

$$\mathbf{r} = \mathcal{R}(h^{(l)}, \mathcal{L}) \tag{7}$$

Additionally, when $|\mathcal{Y}| = v$, we can merge the distribution $\mathbf{p}$ given by $\texttt{LM\_Head}(\cdot)$ and $\mathbf{r}$ given by $\mathcal{R}(\cdot, \mathcal{L})$ with a hyper-parameter $\lambda$:

$$d' = \lambda \cdot \mathbf{r} + (1 - \lambda) \cdot \mathbf{p} \tag{8}$$

which is a common fusion method for mixing two distributions [34; 23; 12; 10].

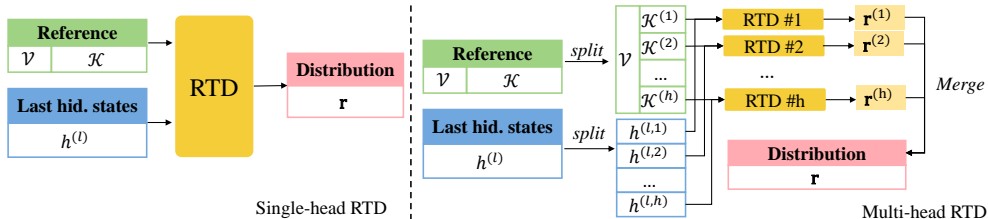

Figure 5: Comparison between RTD and multi-head RTD.

Table 1: Comparison of RTD and MH-RTD on Open Book QA.

| Method | RTD | MH-RTD |
|---|---|---|
| MPT-7B | 27.4 | 30.9 |
| LLaMA2-7B | 47.1 | 52.4 |
| LLaMA2-70B | 63.3 | 65.6 |

### 3.3 Multi-head Reference Trustable Decoding

Large language models like LLaMA2-70B [39] or Mistral-7B [1] utilized MHA and GQA mechanism [2], implies the potential of splitting a large attention vector into smaller ones. So we adapt this method into our RTD process. We define $n_h$ of the head count of the LM model, and $d_h$ the dimension of the each attention head where $d_m = n_h \cdot d_h$. with this in mind, we split the reference datastore into $n_h$ sub-datastore by head. When decoding, we first split $h^{(l)}$ in to heads, then query each sub-datastore and merge the result, showcased in Figure 5. Mathematically,

$$k^{(i)} = k\left[d_h \times (i-1): \ d_h \times i\right], \quad h^{(l,i)} = h^{(l)}\left[d_h \times (i-1): \ d_h \times i\right]$$
$$\mathcal{L}^{(i)} = \{(k^{(i)}, v)|(k, v) \in (\mathcal{K}, \mathcal{V})\}$$

$$(9)$$

And we denote $\mathcal{R}_{\mathrm{MH}}(\cdot, \mathcal{L})$ as the function of the multi-head RTD query process, we have:

$$\mathbf{r} = \mathcal{R}_{\mathrm{MH}}(h^{(l)}, \mathcal{L}) = \frac{1}{n_h} \sum_{i=1}^{n_h} \mathcal{R}(h^{(l,i)}, \mathcal{L}^{(i)})$$

$$(10)$$

### 3.4 Time and Memory Consumption

**Time Consumption**  The time consuming is largely depended on the vector datastore used. For a brute force searching datastore, the time complexity will be $\mathcal{O}\left(s_{\mathcal{L}} \cdot d_m\right)$ where $s_{\mathcal{L}} = |\mathcal{L}|$ is the size of the datastore. However, for those more powerful database like faiss [20] by Meta, with extra training after the generation of reference datastore, the process which have to be done again if the datastore changes, the time consumption can be cut to $\mathcal{O}\left(k \cdot d_m\right)$, where $k$ is a constant related the parameters used to train the database.

For multi-head reference trustable decoding, the performance cost remains the same. The time complexity of each attention-head wise query is $\mathcal{O}\left(d_h \cdot s_{\mathcal{L}}\right)$, the overall query time complexity is $\mathcal{O}\left(n \cdot d_h \cdot s_{\mathcal{L}}\right) = \mathcal{O}\left(d \cdot s_{\mathcal{L}}\right)$, which is the time complexity of convention reference trustable decoding processing. The calculation remains the same for a trained database, the overall time complexity is $\mathcal{O}\left(n \cdot k \cdot d_h\right) = \mathcal{O}\left(k \cdot d_m\right)$.

**Memory Consumption**  The use of time can be optimized by utilizing vector database, however the memory consumption cannot shrink easily. We further define $b$ as the bit cost of the models' `dtype`, where $b_{\texttt{float32}} = 4, b_{\texttt{float16}} = b_{\texttt{bfloat16}} = 2, b_{\texttt{int8}} = 1, b_{\texttt{int4}} = \frac{1}{2}$. The overall memory cost is $d_m \cdot b \cdot s_{\mathcal{L}}$. Due to the lack of lower precision `dtype` support on CPU, even the base model utilized popular half precision `dtype` like `bfloat16`, it still need to be converted into larger ones to be stored. Since all the hidden states have to be saved to calculate precise distanced when rescalled, the memory cost can't be reduced significantly by making it irrelevant with $s_{\mathcal{L}}$.

On the Multi-head RTD side, the memory cost remains the same as the regular RTD takes. The proof is same as the Section 3.4. For instance, reference datastore and head-wise reference datastore with $20,480$ entries with $d_m = 4096$, $n = 32$ and $d_h = 128$, stored in `float32`, takes $320$MB of memory and hard disk space.

**MH-RTD for Resource Saving**  As MH-RTD splits long vectors into multiple smaller ones, it gives us the opportunity to cut time and memory cost by merging different heads together, or directly evict some of them. If on average, $p$ heads are merged into one head, then we expect a $\frac{1}{p}$ resource consumption. The time and memory improvement and corresponding performance impact can be find in the tuning Section 4.3.

# 4  Settings and Experiment

We categorize the common downstream tasks of language models into two types: language understanding and language generation. The former focuses on understanding the input information, based on the context and the information stored within the model, and then outputs the answer in the form of a few tokens, usually in a very simple form. The latter focuses on generating new sentences with complete semantics. We explored the potential of RTD compared to other methods on these two types of tasks. We first compared the effects of RTD and MH-RTD. As shown in Table 1, we found that MH-RTD effectively enhances the capabilities of RTD. Therefore, we default to using the MH-RTD method in the following tests.

## 4.1  Language Understanding

We tested the language understanding capabilities of RTD on multiple benchmarks. When testing, question without answer be shown to the LLM, then we will gather it's baseline output by LLMs' first output token and our RTD result through searching our reference datastore. $\lambda$ is set to $1$ in this task. How the reference datastore is generated can be fount at appendix B.1.

Models we used are: LLaMA2-7B and 70B [39], LLaMA3-8B [8], MPT-7B [37], GLM3-6B [47] [7], Yi-34B. Includes model size from 6B to 70B, as most of the major current models are. We use the *base* version of the model by default. Testing benchmarks are: Massive Multitask Language Understanding (MMLU) [15], AI2 Reasoning Challenge (ARC, both Easy (E) and Challenge (C) parts) [4], Reasoning about Physical Commonsense in Natural Language (PIQA) [5], Open Book Question Answering (OBQA) [30], and Massive Multitask Language Understanding in Chinese (CMMLU) [25]. C-MMLU is a Chinese benchmark, so only Chinese models, GLM3 and Yi, participated in this benchmark.

The multiple-choice benchmarks we chose is challenging enough in itself and requires strong reasoning ability from the model; moreover, the answer format is fixed, which can simultaneously detect the ability to follow instructions. Since that most tasks in the traditional NLP field can be quickly converted into tasks of choosing one from several categories, even some generative tasks, so the results on the multiple-choice test can also represent many other tasks.

The performance boost can be found both with or without ICL. Results are in table 2. Besides testing scores, we also record the confused rate of baseline, the proportion of the questions that failed to be answered properly, including output irrelevant text or can't give a certain answer, in table 3. Meanwhile RTD is designed to given the LLMs' decision in a trustable and controllable way. In comparison with fine-tuning methods in table 4, we can notice that RTD can achieve approximate performance improvements as using PEFT methods like LoRA. Although it is still insufficient compared to full-parameter fine-tuning, the latter has a higher cost and has undergone knowledge injection (which is not considered in this part of the experiment). The dataset used for full-parameter fine-tuning is MMLU-Recall [32; 33], and the hyper-parameters of LoRA can be found in Appendix D. Moreover, we've tested obqa score with different source of reference library, testing the generalization ability of RTD, as shown in Table 5, RTD yields satisfactory results. We've also tested the performance of RTD with different $\lambda$ for language understanding, shown in Table 6. Lastly, we've tested the iteration speed of these benchmarks, as shown in Table 7, the efficiency impact of RTD is minimized comparing to ICL.

| Model | Benchmark | Baseline | 5-shot ICL | RTD ($\Delta$) | 5-shot RTD ($\Delta$) |
|---|---|---|---|---|---|
| LLaMA2-7B | MMLU | 43.8 | 45.8 | 45.1 (1.3↑) | **47.2** (2.1↑) |
| | ARC (E & C) | 30.1 | 65.0 | 41.4 (11.3↑) | **67.3** (2.3↑) |
| | PIQA | 56.5 | 62.1 | 71.4 (14.9↑) | **73.2** (11.1↑) |
| | Openbook QA | 27.8 | 51.0 | 30.4 (2.6↑) | **53.6** (2.6↑) |
| LLaMA2-70B | MMLU | 56.7 | 67.9 | 56.9 (0.2↑) | **68.5** (0.6↑) |
| | ARC (E & C) | 67.4 | 91.6 | 86.1 (19.7↑) | **91.7** (0.1↑) |
| | PIQA | 72.3 | 85.3 | 81.9 (9.6↑) | **86.6** (1.3↑) |
| | OpenbookQA | 53.7 | 84.4 | 68.2 (14.5↑) | **85.4** (1.0↑) |
| LLaMA3-8B | MMLU | 47.5 | **63.9** | 57.2 (9.7↑) | 61.9 (2.0↓) |
| | ARC (E & C) | 71.2 | **87.3** | 83.7 (12.5↑) | 87.1 (0.2↓) |
| | PIQA | 69.9 | 78.9 | 76.3 (6.4↑) | **80.0** (1.1↑) |
| | OpenbookQA | 53.3 | 77.5 | 71.4 (18.1↑) | **78.6** (1.1↑) |
| MPT-7B | MMLU | 27.4 | 29.6 | **30.4** (3.0↑) | 29.8 (0.2↑) |
| | ARC (E & C) | 27.5 | failed | 27.6 (0.1↑) | **30.1** |
| | OpenbookQA | 29.4 | failed | 27.2 (2.2↓) | **30.4** |
| GLM3-6B | MMLU | 41.9 | 48.6 | 47.6 (5.7↑) | **49.8** (1.2↑) |
| | ARC (E & C) | 59.1 | 75.3 | 75.0 (15.9↑) | **76.5** (1.2↑) |
| | PIQA | 66.8 | 73.6 | **75.9** (9.1↑) | 74.5 (0.9↑) |
| | OpenbookQA | 55.1 | 67.1 | 64.0 (8.9↑) | **68.8** (1.7↑) |
| | C-MMLU | 48.8 | 54.5 | 53.3 (4.5↑) | **54.7** (0.2↑) |
| Yi-34B | MMLU | 68.6 | **74.3** | 70.3 (1.7↑) | 73.3 (1.0↓) |
| | ARC (E & C) | 93.3 | 94.0 | 90.7 (2.6↓) | **94.6** (0.6↑) |
| | PIQA | 88.3 | 83.5 | **88.4** (0.1↑) | 87.7 (4.2↑) |
| | OpenbookQA | 83.5 | **89.8** | 88.4 (0.9↑) | 88.8 (1.0↓) |
| | C-MMLU | 70.3 | 81.0 | 73.9 (3.6↑) | **81.8** (0.8↑) |
| **Avg** | - | 56.41 | 65.28 | 63.31 | **68.88** |

Table 2: RTD on language understanding benches. Baseline refers to zero-shot performance. ICL exceeds MPT-7B's 2048 context window, with a 0 score result, recorded as failed in the table.

Table 3: Confused rate.

| Model | Llama2-7B | GLM3-6B | Yi-34B |
|---|---|---|---|
| Rate | 8.6% | 11.81% | 0.44% |

Table 4: RTD comparing with fine-tune methods.

| Methods | baseline | LoRA | FT | RTD |
|---|---|---|---|---|
| Score | 41.9 | 42.5 | 46.31 | 42.8 |

## 4.2 Language Generation

**Reasoning with Context** Generative tasks are generally subjective and difficult to test. We constructed a benchmark based on Retrieval-Augmented Generation (RAG) and Open Book Question Answering [30] to test the potential of RTD in areas requires advance reasoning such as knowledge injection. Chain-of-Thought [42] is a method that encourage the model to provide a step-by-step analysis before giving the final answer, thereby enhancing the model's capabilities. We compared the performance of the model when introducing references through the ICL method and the RTD method, to determine the effectiveness of the RTD method. The extra knowledge source was Wikipedia. The generation of the datastore can be found in detailed in Appendix B.2. With the results of table 8, it can be seen that RTD was indeed helpful in knowledge injection. Besides, the context length is shrunk by a lot, thus saves reasoning GPU time and memory consumption. A detailed exploration of why RAG score is lower than baseline can be find in Appendix C.

**Style transfer** To explore whether the RTD method can be used to modify the language style of the model, we designed a style transfer experiment. We used a moderately scaled and strongly styled dataset, Tiny-Shakespeare [21; 40], and compared the perplexity (PPL) of the model on the test set after LoRA and RTD, to measure whether our method can help the model change the output style.

<table>
<tr><td colspan="4">Table 5: Generalization of RTD.</td></tr>
</table>

| Source | OBQA | ARC | MMLU |
|---|---|---|---|
| OBQA | 71.4 | 71.4 | 71.2 |

Table 6: Different $\lambda$ in Language Understanding

| $\lambda$ | 1 | 0.8 | 0.6 | 0.4 | 0.2 | 0 |
|---|---|---|---|---|---|---|
| OBQA | 71.4 | 68.0 | 67.0 | 66.8 | 66.6 | 53.3 |

Table 7: Efficiency of RTD.

| Methods | baseline | RTD | ICL | ICL + RTD |
|---|---|---|---|---|
| Speed(it/s) | 25.1 | 23.6 | 7.90 | 7.85 |
| Extra Memory Usage (MB) | 0 | ~16 | ~37 | ~52 |

The results in Table 9 prove that our RTD method can reduce the perplexity of the model, enabling the model to adapt to the style of different datasets. The hyperparameters of LoRA are in Appendix D.

### 4.3 Influence of Hyper-parameters in RTD

Although our method is quick and efficient, it still introduces several hyper-parameters. We hope to explore the relationship between these hyper-parameters and the final performance of RTD. We conducted a series of ablation experiments on LLaMA2-7B [39] and OBQA [30] to explore the impact of different hyper-parameters on performance and how to quickly determine the optimal hyper-parameters. The overall result can be found in Figure 6. If not tuned, we set $k = 1024$, $s_{\mathcal{L}} = 19,828$, $\lambda = 1$ and $T = 750$ by default.

Depicted in Figure 6 (a), RTD's performance improves initially with increasing $s_{\mathcal{L}}$ but eventually maxed out and starts oscillating when $s_{\mathcal{L}}$ reaches 4096. Generally, a larger $s_{\mathcal{L}}$ gives a better performance, but it do get maxed out depends on the specific task. Figure 6 (b) showcased us how RTD's performance consistently improves as $k$ increases initially, but eventually reaches a plateau, similiar with the $s_{\mathcal{L}}$. To be denoted is that a larger $k$ could harm efficiency. Figure 6 (c) implies that RTD can only reach it's best performance when $T$ is large enough. Empirically, due to the characteristics of the exponential function, as long as the range of scaled distances $d''$ is kept between 1-2, a sufficiently good effect can be achieved. In RTD, $\lambda$ is an important variable, especially in generation tasks. However, $\lambda$ does not require high precision, and the range is relatively limited, so a good enough effect can be achieved quickly through a few attempts. Empirically speaking, 0.4-0.7 is a suitable range for $\lambda$. Previous studies indicated that by pruning the dimension of attention won't hurt $k$nn algorithm's performance [13]. In the case of RTD, showcased in Figure 6 (d), it can be found that the performance won't drop with at least $\frac{1}{4}$ heads remained, and the generation speed was boosted as more heads are dropped.

## 5 Conclusions

In this paper, we introduce Reference Trustable Decoding, a novel training-free method designed to augment Large Language Models in downstream tasks. RTD refines the output distribution by leveraging references retrieved from a specially curated datastore, as a bypass of conventional `LM_Head`. Our experimental results demonstrate RTD achieved superior performance compared to the In-Context Learning baseline in 21 out of 25 different dataset and model configurations as well as fine-tune based methods. This result highlights the effectiveness of RTD across a diverse range of scenarios, underscoring its potential as a robust solution for enhancing language model capabilities in downstream tasks.

## Limitations & Future Work

RTD is an efficient and quick method to augment the capabilities of models on specific downstream tasks. However, for some tasks, especially generative tasks, the large reference datastores that are difficult to directly compress may pose challenges for applications. Nevertheless, we believe that there is likely inherent redundancy in such large datastores. We hope to enable machines to identify these redundancies while maintaining a gradient-free method, in order to achieve efficient fine-tuning.

Table 8: Comparison of RTD and RAG using Wikipedia on LLaMA2-7B-Chat.

| LLaMA2-7B-Chat | Acc | Latency (ms) |
| --- | --- | --- |
| Baseline | 39.0 | **42.5** |
| Wiki RAG | 29.0 | > 200 |
| Wiki RTD | **44.4** | 46.5 |

Table 9: PPL of the fitted model on domain datasets.

| Dataset | Baseline | LoRA | RTD |
| --- | --- | --- | --- |
| Tiny-S | 1.6982 | 1.3710 | 1.4501 |

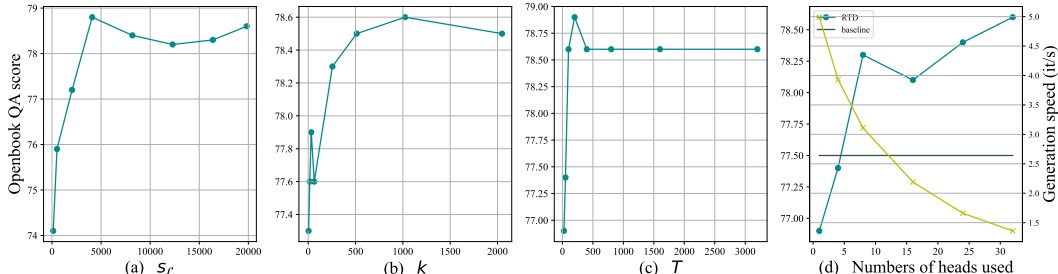

Figure 6: Hyper-parameters' influence on RTD's performance

How to make RTD accomplish tasks with high quality while being space-efficient is our following research direction.

## Acknowledgments

We sincerely appreciate the valuable feedback provided by all reviewers during the review process, as well as the efforts of the area chairs. This work was supported by the National Natural Science Foundation of China (No. 62306216), the Natural Science Foundation of Hubei Province of China (No. 2023AFB816), the Fundamental Research Funds for the Central Universities (No. 2042023kf0133).

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

# Appendix

## A   Testing Environments

All testing are done on a server with 8*A100 80G SXM. For models with less than 15B parameters, 2 of 8 GPUs are used. For models with more than 15B parameters, 4 of 8 GPUs are used. All testing are carried out under HuggingFace Transformers library [43].

## B   Generation of Reference Datastore

### B.1   Benchmark Testing

To generate reference datastores, LLMs are shown to the questions and options in the training split of the benchmarks and we store the attention output. For each question this process is repeated four times cycling through A, B, C, D as the correct value. $3,500$ to $5,000$ question is shown to the LLMs and about $20,000$ $(k,v)$ entries are generated. To be noted is that the reference datastore of CMMLU is generated from validation set of C-eval [17], split *zho_Hans* of belebele [3] and testing set of ACLUE [49] since there is no training split for the benchmark.

### B.2   Wikipedia Fact Retrieval

For our reference datastore, we encoded all of the Wikipedia sentences using the Jina [11] model, which is smaller in both it's parameter count and hidden size, resulting in a faster generation speed and smaller space cost for encoded vector datastore. Every usable sentence in Wikipedia is encoded, meanwhile the sentences from the same page share a same value, which is the no. of this page. When testing, we use the same model to encode the question, then we search the most relevant pages in the datastore, be the metric of cosine similarity, to retrieve the most relevant pages. In this section, $s_{\mathcal{L}}$ is same as the sentences count of Wikipedia, around 73M. $k = 1024$. $T$ and $\lambda$ are not applicable here.

With retrieved pages, we generate a reference datastore with every sentence in the pages. We first calculate attention representations for every token, whose corresponding value is the id of next token, *eos* for the last token. Then we use this dynamically generated reference datastore for following RTD. In this section, $s_{\mathcal{L}}$ is the same as the length of tokenized sequence, 6200 on average, and we use $T = 750, k = 1024, \lambda = 0.4$.

## C   RAG's Deficiency in Testing

RAG method's shows a decline in performance in Table 8. To explain this, we can further examine the average length of the tokenized sequences of the retrieved context, which is around 6200, showcased in Table 10. This length will hardly increase any inference cost for the RTD method, due to the small $s_{\mathcal{L}}$, but it exceeds the pre-training sequence length of LLaMA2-7B-Chat, which is 4096. That is to say, the naive RAG method here will cause sequence length overflow, thereby significantly affecting performance. If the overflow happened, then the model's ability is cut down significantly.

## D   LoRA Hyperparamters

See Table 11. For LoRA tuning on MMLU, any question whoes tokenized length exceed 4096 was evicted from both training and testing. The maximum tokenized length of the Tiny-Shakespeare dataset is 900.

Table 10: Average length by token in OBQA question answering process, split by sections.

| Section | Average Length |
|---|---|
| Wikipedia Context | 6192 |
| Question | 84 |
| Response | 231 |

Table 11: LoRA Hyper-parameters

| Hyper-parameter | Value |
| --- | --- |
| Batch Size | 4 |
| Epochs | 2 |
| Max Seq. Len. | 4096 |
| LoRA Target | {Q, K, V, O, Up, Down, Gate}_proj |
| LoRA Rank | 16 |
| LoRA $\alpha$ | 32 |
| LoRA dropout | 0.01 |
| Learning Rate | 1e-5 |
| Optimizer | AdamW |
| Adma RMS $\epsilon$ | 2e-4 |
| Adam $\beta$ | $(0.9, 0.999)$ |
| Adam Weight Decay | 0.01 |
| Scheduler | Constant LR |

