# OpenReview forum: "Reference Trustable Decoding: A Training-Free Augmentation Paradigm for Large Language Models"
_NeurIPS.cc/2024/Conference — NeurIPS 2024 poster_

### Official Review · Reviewer_tphZ · 2024-06-22

**Soundness:** 3
**Presentation:** 3
**Contribution:** 3
**Rating:** 6
**Confidence:** 4

**Summary:**

This paper proposes Reference Trustable Decoding (RTD), a new paradigm that allows models to adapt to new tasks without fine-tuning and has lower inference costs compared to in-context learning (ICL). RTD uses the last-hidden states of the input sample to retrieve similar samples in the training examples and optimizes the LLM’s final vocabulary distribution using these examples’ last-hidden states. Experimental results on various LLMs show RTD can effectively adapt LLM and it can be used together with ICL method to further boost task performance.

**Strengths:**

1. The proposed method is well-motivated, addressing the weaknesses of ICL and fine-tuning in model adaptation. The paper is also clearly written, with Figure 2 providing a clear comparison of RTD and baseline methods.
2. The experimental results on multiple task setups showcase the effectiveness of the RTD in adapting the model. Also, the proposed method can be used together with traditional methods to further boost performance.
3. Compared to RAG or ICL, RTD selects relevant examples based on the similarity in the final hidden state space. Many existing techniques for vector database can be used to boost the efficiency of RTD.

**Weaknesses:**

1. The description of using RTD in language generation task is unclear. When generating a sequence of multiple tokens in an autoregressive way, how does the retrieved sample adapt the vocabulary distribution after the first token? Also, it is unclear what the constructed “benchmark based on RAG and Open Book Question Answering” in Section 4.2 is.
2. The performance gain of RTD seems to be unstable as Table 2 shows on some datasets, RTD is much worse than 5-shot ICL. The paper lacks discussion on what tasks is RTD suitable to use.
3. The paper should have a part clearly describe the baseline methods in comparison. How are the 5 examples selected in “5-shot ICL”? I think this is quite important, as some previous works use similarity in the sentence embedding space to select in-context examples to adapt the model via ICL. Given the proposed method mainly leverage the last hidden state space, more careful comparison and discussion can help understand whether it’s more helpful to leverage the last hidden state space.

**Questions:**

See my questions in “Weaknesses”.

**Limitations:**

Yes. It is discussed in "Limitations & Future Work".

---

> ### Author Rebuttal · Authors · 2024-08-06
>
> We sincerely appreciate your willingness to thoroughly read our paper. The time you’ve invested in our work is truly invaluable to us. Your recognition of our method’s clear motivation, extensive experiments, and ease of optimization is an honor. Additionally, we thank you for pointing out the shortcomings in our paper. We highly value your feedback, and these suggestions will undoubtedly contribute to refining our work.
>
> ### Weakness
>
> 1. **How RTD behave when auto-regressive decoding:**
>     Regarding RTD, it follows the traditional autoregressive decoding pattern, where the generation of subsequent tokens is handled in the same manner as the first token. Specifically, after generating a token, it is appended to the input sequence, and the new last hidden states are used to perform another round of RTD for the next token. We will provide more detailed explanations in future versions of the paper to clarify these points.
>
> 2. **What is the benchmark based on RAG and Open Book Question Answering:**
>   For benchmark based on RAG and Open Book Question Answering, we add two components to the original QA setup to test whether RTD could inject new knowledge into LLMs:
>
>     1. Source of New Knowledge:
>       We chose Wikipedia articles, and the filter the relevant pages out using a RAG system.
>     2. Process of Knowledge Injection:
>       Since RTD operates at the token level, and Wikipedia articles do not directly contain options like A, B, C, and D, we need a process where the model generates an analysis of the question. During this generation process, RTD injects new knowledge. So we adopted a CoT  decoding approach for this purpose.
>
>    We will provide more detailed descriptions in the revised version.
>
> 3. **The unstable performance of RTD in some dataset:**
>   In the NLU experiments, whether it’s fine-tuning, ICL, or our RTD approach, the core goal is not merely injecting more knowledge into the model but rather guiding the model to better utilize its existing knowledge. Essentially, any optimization method is constrained by the upper limit of the model’s knowledge capacity. Breaking through this limit is not feasible given the current computational budget. Even if these methods are orthogonal, when the model has already touched or approaching this upper limit, further improvement in capability becomes restricted. In some cases, fluctuations due to errors may occur. Our primary experiments have demonstrated two key points:
>
>     1. RTD alone significantly enhances model performance
>     2. RTD has the potential to complement methods like ICL
>        However, when ICL is already sufficient to reach the theoretical upper limit, combining it with RTD may unavoidably introduce performance variations due to error factors.
>
>    We will add more detailed analysis in the next version.
>
> 4. **Detailed description to baseline:**
>   Thank you for your suggestion. We will provide a more detailed discussion in future version. For most benchmarks, specific example data is already prepared for constructing ICL. For instance, in the case of the MMLU benchmark, the *dev* subset provides five sample questions for each question category. This is why we chose to use a 5-shot setup. For benchmarks that do not have corresponding sections explicitly designed for ICL, we randomly sample five questions from the provided training data to create a 5-shot setup. Both the test group and the control group use the same set of questions to ensure experimental fairness.
>
> 5. **Discussion about the Last Hidden Space:**
>   For last hidden space, our modification target lies within the *LM_Head*, and the Last Hidden State serves as the input to the *LM_Head*. The *LM_Head* represents the final layer of the entire LLM network, which means that our method essentially operates on a relatively shallow portion of a deep neural network. This design choice provides us with an opportunity to employ richer methods compared to a single gradient descent approach. Additionally, it allows us to maintain a “training free” approach. In fact, the core of our paper lies in exploring the optimization of the model using the Last Hidden State. Our ultimate goal is not only to bypass the *LM_Head* but also to delve further into the properties of this specific region. Notably, in some newer models like llama3-8b, the *LM_head* constitutes a substantial portion of the model parameters (6.6% or 0.53B out of 8.03B), yet it has received relatively little attention.
>
> Finally, we would like to express our gratitude for the time you’ve spent reviewing our paper again. Your suggestions have been highly relevant to the content and have significantly contributed to improving the paper. If you have any further questions or additional feedback, please feel free to reach out, and we will promptly respond.

---

> > ### Comment · Reviewer_tphZ · 2024-08-10
> >
> > Thanks for the detailed response!
> >
> > > RTD has the potential to complement methods like ICL However, when ICL is already sufficient to reach the theoretical upper limit, combining it with RTD may unavoidably introduce performance variations due to error factors.
> >
> > I think that's why I have the question "The paper lacks discussion on what tasks is RTD suitable to use". Given the current explanation, shall I always use RTD on all my tasks?

---

> ### Author Response · Authors · 2024-08-10
>
> **Dear reviwer tphZ**,
>
> We sincerely appreciate your timly reply.
>
> We propose RTD, an approach that provide a new balance between performance and efficiency. We conducted supplementary tests on the *OBQA* benchmark, with *LLaMA-3-8B*, and $20,000$ RTD entries, comparing different methods in terms of generation speed and memory footprint. Notably, RTD exhibits significant advantages over ICL and performs on par with the baseline in the aspact of generation speed. Analyzing memory usage, RTD’s additional memory footprint is only $\frac{1}{2}$ of ICL’s, resulting in reduced bandwidth bottlenecks and computational load. For other datasets, especially those with longer contexts, RTD’s efficiency advantage becomes even more pronounced, as ICL’s extra overhead depends on context length, whereas RTD remains context-agnostic.
>
> | Methods | Speed (it/s) | Extra Memory Usage (MB) |
> |-|-|-|
> | Baseline (PEFT) | $25.1$ | $0$ |
> | RTD | $23.6$ | ~$16$ |
> | ICL | $7.90$ | ~$37$ |
> | ICL + RTD | $7.85$ | ~$52$ |
>
> Considering both performance and efficiency analysis, **we can conclude that RTD offers a satisfactory solution for various task types due to its excellent cost-effectiveness, which makes it always worth trying**. If RTD alone doesn’t fully meet your performance requirements and you’re willing to sacrifice some efficiency, consider using the ICL method to assess performance. Alternatively, combining ICL with RTD may yield a further performance boost without significantly compromising efficiency in most tasks. Finally, if you have abundant computational resources and performance remains unsatisfactory, fine-tuning could be explored. A prefered workflow with RTD can be look like:
>
> ``` verilog
> prompt tuning -> RTD -> ICL -> ICL+RTD -> finetune
> ```
>
> From left to right both the cost and expected performance increase. And this workflow demonstrates how RTD seek a new balance.
>
> In our upcoming revisions, we will include more comprehensive efficiency tests and further analyze the trade-off between efficiency and performance of variouse methods, emphasizing the high efficiency of our proposed method.
>
> **Best regards,**
>
> **Authors**

---

> > ### Comment · Reviewer_tphZ · 2024-08-11
> >
> > I really thank the authors for providing the additional results and the preferred workflow. I think adding this discussion to the paper could be helpful and could potentially increase the adoption of the proposed method.

---

> > > ### Author Response · Authors · 2024-08-12
> > >
> > > We sincerely appreciate your positive feedback on our work. Your suggestions have indeed helped us improve the paper and address crucial aspects. We will ensure to incorporate these additions in the next version. Once again, thank you for your time and effort; they are invaluable to us.

---

### Official Review · Reviewer_vZ3x · 2024-07-13

**Soundness:** 3
**Presentation:** 2
**Contribution:** 2
**Rating:** 4
**Confidence:** 3

**Summary:**

The paper introduces Reference Trustable Decoding (RTD), a novel framework that allows large language models to adapt to new tasks without the need for fine-tuning. RTD constructs a reference datastore from training examples and optimizes the LLM’s final vocabulary distribution by selecting suitable references based on the input. This results in more reliable responses and low-cost adaptation to downstream tasks. The paper includes experimental evaluations demonstrating RTD's effectiveness across various benchmarks and tasks, highlighting its potential as a new paradigm for augmenting LLMs.

**Strengths:**

1. The paper presents a unique and training-free method for augmenting LLMs, which is a significant contribution to the field.
2. The methodology is clearly explained, and the figures provided help in visualizing the RTD process.
3. Comprehensive experimental evaluations are conducted, demonstrating the effectiveness of RTD across different tasks and benchmarks.
4. The paper discusses the orthogonality of RTD with traditional methods, indicating its potential for combined usage and enhanced performance.
5. The authors provide a thorough analysis of the hyper-parameters involved in RTD and their impact on performance.

**Weaknesses:**

1. While the paper discusses the efficiency of RTD, it does not provide a direct comparison with other methods in terms of computational resources or scalability.
2. The paper could benefit from a more detailed discussion on the potential applications of RTD in real-world scenarios.
3. The experimental section, although comprehensive, might have benefited from additional benchmarks, particularly those that test the generative capabilities of LLMs further.
4. The paper mentions the inability to compress large reference datastores as a challenge; however, it does not provide insights or potential solutions to this issue.

**Questions:**

1. How does RTD compare with other state-of-the-art methods in terms of computational efficiency and scalability, especially for larger datasets or more complex tasks?
2. Can the authors provide more details on how RTD might be integrated into existing systems or workflows where LLMs are already deployed?
3. The paper mentions the potential for RTD to be combined with other methods like ICL and fine-tuning. Are there specific strategies or best practices for such integration?

**Limitations:**

1. The paper acknowledges the challenge of compressing large reference datastores but does not offer solutions or workarounds for this limitation.
2. The scope of the experiments could be expanded to include a broader range of tasks, especially those that are more generative in nature.
3. While the paper discusses the performance of RTD in a controlled setting, it does not address how well these results might generalize to other domains or tasks outside the tested benchmarks.
4. The paper does not provide a detailed discussion on the ethical considerations or societal impacts of deploying LLMs with enhanced capabilities through RTD, which could be important for responsible AI development.

---

> ### Author Rebuttal · Authors · 2024-08-06
>
> We sincerely appreciate the time you’ve spent on our paper. Thank you for recognizing our approach, experiments, and writing. The following is our response regarding the weaknesses, questions, and limitations, you raised. We aim to clarify any potential misunderstandings and indicate the direction of modifications for our paper.
>
> ### Weakness
>
> 1. **Comparison with other methods:**
>   In the second section of our paper, we discussed the comparison between our approach and other methods in terms of efficiency and capability. However, we plan to supplement a small table to provide a concise and quick overview. The table would look like this:
>
>    | Methods | Training Cost | Inference Cost | Capability |
>    |-|-|-|-|
>    | No Method | Low | Low | Low |
>    | RTD (Ours) | Minimum | Minimum | Moderate |
>    | PEFT | High | Low | Moderate |
>    | ICL | Low | High | Moderate |
>    | FFT | Gigantic | Low | Best |
>
>    And we've also tested the inference efficiency of RTD and it's counterparts.
>
>    | Methods | Speed (it/s) |
>    |-|-|
>    | RTD | $23.6$ |
>    | PEFT | $25.1$ |
>    | ICL | $7.9$ |
>
> 2. **Real-world application:**
>   In our experiments, we utilized highly challenging and widely used datasets that are derived from real-world scenarios, such as MMLU which covers 57 subjects across STEM, the humanities, the social sciences, and tests both world knowledge and problem-solving ability. This makes the benchmark more challenging and more similar to how we evaluate humans. We will provide more detailed case studies in future versions of the paper.
>
> 3. **Additional generative benchmarks:**
>   In section 4.2, we tested different LLM generation tasks, including reasoning with context and style transfer tasks. For example, we used a Retrieval-Augmented Generation (RAG) benchmark to evaluate knowledge injection and the ability to provide step-by-step reasoning using Chain-of-Thought prompting. Additionally, we conducted style transfer experiments using the Tiny-Shakespeare dataset to measure the model's ability to adapt to different language styles.
>   In our revised version, we will include additional examples of RTD-generated instances to support the effectiveness of the RTD method.
>
> 4. **Solutions of large datastores:**
>   While we acknowledge the importance of compressing large reference datastores, this particular challenge falls outside the primary scope of our current research. Our main focus was on introducing the reference trustable framework (RTD), which aims to better integrate and utilize reference datastores.
>   Besides, in Sections 3.4 and 4.3 of our paper, we discussed a resource-saving approach by reducing the number of utilized heads, which can partially solve this.
>   We see this limitation as an opportunity for future research rather than a weakness of our current work. We believe RTD provides a solid foundation for further exploration into effective compression techniques. We appreciate your insight and will consider discussing potential approaches in future works.
>
> ### Questions
>
> 1. **Larger and more complex tasks:**
>   We would like to emphasize that the dataset used in our experiments in Section 4, such as *MMLU*, is already a large-scale and challenging benchmark commonly used for testing. Its complexity and difficulty level are significant.
>
> 2. **Integrated into existing systems:**
>   This aspect was indeed underrepresented in our paper, and we agree that it deserves further elaboration. For instance, when using LLM for text screening, a small set of screening cases can be used to generate a Reference Datastore. By combining this with RTD, we can replace the original ICL portion, thereby increasing screening efficiency and reducing overall computational load. Additionally, analyzing screening results can help reverse-engineer potential issues in the Reference Datastore, highlighting the “Trustable” feature of RTD.
>
> 3. **Combine methods**
>   In Table 2, we have conducted experiments combining RTD with other methods, demonstrating that generally, the more methods we combine, the better the overall performance tends to be. However, such improvements often come with additional computational resource requirements, necessitating a trade-off between performance and cost.
>   **The best integration strategy needs to be determined based on the specific use case**. The core contribution of RTD lies in providing a method to augment model capabilities without imposing **excessive additional computation**, thus opening up new possibilities for model enhancement under limited resources.
>
> ### Limitations
>
> 1. **Large datastores:**
>   See weakness #4
>
> 2. **Generative testings:**
>   See weakness #3
>
> 3. **Settings generalization:**
>   RTD has been tested on two types of tasks: language understanding and language generation, across a total of **seven different benchmarks** and **six models**. We believe this extensive evaluation effectively demonstrates RTD's ability to generalize to various domains and tasks.
>
> 4. **Responsible AI development:**
>   We propose RTD as a model enhancement method, which will have similar societal effects as other existing methods and won’t introduce any additional issues for discussion. We will emphasize this point more clearly in future versions of the paper.

---

> ### Author Response · Authors · 2024-08-12
>
> **Dear Reviewer**,
>
> We sincerely appreciate your thorough review and valuable feedback. We have taken your comments seriously and have already conducted additional experiments and provided further explanatory details based on your recommendations.
>
> We've further measured our RTD methods on efficiency, both time and memory comsumption wise, here is our latest result.
>
> | Methods | Speed (it/s) | Extra Memory Usage (MB) |
> |-|-|-|
> | Baseline (PEFT) | $25.1$ | $0$ |
> | RTD | $23.6$ | ~$16$ |
> | ICL | $7.90$ | ~$37$ |
> | ICL + RTD | $7.85$ | ~$52$ |
>
> Considering both performance and efficiency analysis, **we can conclude that RTD offers a satisfactory solution for various task types due to its excellent cost-effectiveness, which makes it always worth trying**. A prefered workflow with RTD can be like:
> ``` verilog
> prompt tuning -> RTD -> ICL -> ICL+RTD -> finetune
> ```
>
> To ensure that we address the issues you raised effectively, we kindly request your prompt response. Your feedback is crucial for enhancing our work. We understand that the review process is demanding, and we greatly appreciate your time and effort.
>
> Thank you once a gain for your support and help.
>
> **Best regards,**
>
> **Authors**

---

### Official Review · Reviewer_wxwA · 2024-07-13

**Soundness:** 2
**Presentation:** 2
**Contribution:** 2
**Rating:** 5
**Confidence:** 3

**Summary:**

Adapting large language models (LLMs) to specific tasks remains costly and complex. In-context learning (ICL) and parameter-efficient fine-tuning (PEFT) still suffer from inference latency and training costs, respectively. To address these problems, this paper proposes Reference Trustable Decoding (RTD), a training-free paradigm for better adapting base LLMs to downstream tasks. RTD is a retrieval-based framework that creates a reference datastore from task datasets, storing key-value pairs of hidden states and corresponding labels. During inference, RTD retrieves the most relevant references based on the hidden states of the input and then aggregates them with model logits to compute the final output distribution. Experimental results demonstrate that RTD effectively adapts mainstream LLMs for natural language understanding (NLU) and natural language generation (NLG) benchmarks, outperforming ICL approaches.

**Strengths:**

1. **Efficient Adaptation:** The proposed RTD method integrates external knowledge and constraints to adjust outputs without increasing input length or using gradient descent.
2. **Orthogonality:** As a plug-and-play decoding paradigm, RTD can be flexibly integrated with existing techniques such as ICL and fine-tuning.
3. **Significance:** The paper demonstrates the excellent performance of the RTD method on multiple benchmarks, surpassing ICL methods and baselines. The RTD method is particularly meaningful in scenarios requiring rapid iteration of content or controllable generation.

**Weaknesses:**

1. **Related Work:** As an RTD method based on retrieval, the experiment sections should discuss and compare it with similar studies such as kNN-LM [1] (which also is cited in the paper) and other retrieval-based decoding methods.
2. **Generalization Analysis:** The proposed method relies on the construction of a reference datastore, however, the discussion on out-of-domain inputs is still lacking. This poses a potential risk to the generalization of RTD method.
3. **Hyperparameter Challenges:** Ablation experiments show that the effectiveness of RTD relies on adjusting several hyperparameters, which may hinder the method's scalability. In practical use, applying the RTD method to new models and tasks may require additional hyperparameter searches on a held-out validation set.
4. **Trustable Decoding Association:** The discussion in the paper may not be strongly associated with "Trustable Decoding" [2]. Utilizing a reference datastore provides transparent reference data for base model decoding, achieving in-domain controllable generation. However, the core argument of the paper still emphasizes efficiently transferring base LLMs using training-free methods.
5. **Typos:**
   - Line 242: "Detailed" should be "detail."

References:

[1] Khandelwal, U., Levy, O., Jurafsky, D., Zettlemoyer, L., & Lewis, M. (2020). Generalization through Memorization: Nearest Neighbor Language Models. In International Conference on Learning Representations.

[2] Liu, Y., Yao, Y., Ton, J. F., Zhang, X., Cheng, R. G. H., Klochkov, Y., ... & Li, H. (2023). Trustworthy LLMs: A survey and guideline for evaluating large language models' alignment. arXiv preprint arXiv:2308.05374.

**Questions:**

In NLU tasks, why is the value of $\lambda$ set to 1? Under this setting, the distribution of the output tokens entirely comes from RTD, without integrating the outputs from the LM head itself. Lines 268-269 also mention that the value of $\lambda$ empirically ranges between 0.4-0.7. Setting it to 1 might require some justification or explanation in the paper.

**Limitations:**

The authors have included Limitation section after conclusion.

---

> ### Author Rebuttal · Authors · 2024-08-06
>
> We sincerely appreciate your thorough reading of our paper. The time you’ve invested in our work is truly invaluable to us. Your endorsement of the efficiency, orthogonality, and practical significance of our proposed RTD method means a great deal, and we are genuinely grateful for your recognition.
>
> Additionally, we thank you for pointing out the limitations and constraints of RTD. Such constructive feedback is immensely helpful in improving our paper.
>
> ### Weakness
>
> 1. **Lack of comparison with other retrieval-based decoding methods:**
>   The reason we didn’t compare with retrieval methods lies in two factors. First, other retrieval-based methods typically rely on an gigantic reference datastore. For instance, in the original paper of $k$NN-LM, over 1 billion entries were used, which significantly exceeds the data volume typically employed in our RTD method (usually less than 1 million). This disparity makes direct comparisons a lower reference value, especially given RTD’s focus on efficiency. Second, running a complete $k$NN-LM requires substantial hardware resources, often involving massive CPU memory and potentially distributed cluster deployments. Unfortunately, we lack the necessary resources to fully replicate knn-LM’s performance.
>
> 2. **Lack of generalization analysis:**
>   Thank you for your suggestion. We want to emphasize that RTD’s core objective is to efficiently augment LLMs within specific domains. However, exploring cross-domain generalization is indeed a promising avenue. We plan to supplement our paper with cross-domain tests within the NLU field, given the structural similarities across different benchmarks. Preliminary exploration suggests that our RTD method exhibits some cross-domain generalization capability, particularly when task formats are similar. As soon as detailed testing is complete, we’ll promptly share the results. Our first batch of result are as follows, we tested the reference datastore generate from different benchmarks on OBQA.
>
>   |Datasore source|OBQA|ARC|MMLU|
>   |-|-|-|-|
>   |score|$71.4$|$71.4$|$71.2$|
>
> 3. **Requirement on additional hyperparameter searches:**
>   Our method introduces a certain number of hyperparameters, and determining the optimal hyperparameters does indeed pose new challenges for RTD. However, in Section 4.3 of our paper, we extensively discuss the impact of hyperparameters on performance. Our experiments reveal that RTD is relatively insensitive to hyperparameters, and performance trends are quite evident as hyperparameters vary. Therefore, while identifying an optimal set of hyperparameters still requires some experimentation, finding a satisfactory set is relatively simple and straightforward.
>
> 4. **Lacks association with trustable decoding:**
>   Thank you for your suggestion. Our proposed RTD method fundamentally targets the *LM_Head* within the entire LLM network. Since the *LM_Head* is the final layer of the LLM, our method essentially operates on a relatively shallow portion of a deep neural network. This design choice allows us to employ more interpretable methods compared to traditional gradient-based optimization.
>   For example, if during RTD deployment, we observe unexpected model behavior or undesirable outputs, we can rectify these issues by inspecting the data within the Reference Datastore. The data in the Reference Datastore is traceable, meaning we can identify which specific example led to an error and easily remove it. In contrast, achieving a similar process with gradient-based methods, such as gradient descent, would be extremely challenging, if possible at all.
>   We will supplement relevant content in the next version to address this issue and further clarify the association with "Trustable Decoding."
>
> 5. **Typos:**
>   We are sorry that there are some typos in our paper, we will correct them immediately. Thank you for your careful reading.
>
> ### Questions
>
> - **The explainaion of setting $\lambda$ to 1:**
>   In RTD, we introduce a hyperparameter $λ$ to blend the results from RTD and those from the base model. The rationale behind this choice is that we typically don’t want to completely discard information from the base model. By doing so, we avoid the risk of RTD inadvertently causing a regression in overall system performance, ensuring greater robustness. In the specific case you mentioned, which pertains to NLU tasks, the relatively constrained format allows RTD to perform well on the task itself while enhancing performance. In such scenarios, we believe that additional information from the base model is unnecessary. Furthermore, setting $λ$ to $1$ avoids costly vocabulary operations, conserving computational resources, especially crucial for large-vocabulary modern models like llama3-8b, where the *LM_head* constitutes 6.6% of the model parameters (0.53B out of 8.03B). We appreciate your perspective, and indeed, the setting of $λ$ still worth exploration for this task. We will conduct an additional experiment to present the influence of $λ$. We will provide more detailed discussion in the next version.
>   Our result of tuning $\lambda$ on LLaMA3-8B, OBQA are as follows:
>
>   |$\lambda$|$1$|$0.8$|$0.6$|$0.4$|$0.2$|$0$|
>   |-|-|-|-|-|-|-|
>   |score|$71.4$|$68.0$|$67.0$|$66.8$|$66.6$|$53.3$|
>
>
> Allow us to express our gratitude once again for your diligent reading of our paper. Your suggestions are valuable and can definitely help us to improve our paper. If you have any further insights or questions, please share them with us.

---

> > ### Comment · Reviewer_wxwA · 2024-08-13
> >
> > Dear Authors,
> >
> > Thank you for your response. Previously, I have acknowledged the efficiency of this work in the original review. Did you accidentally place the efficiency results in the wrong comment box?
> >
> > I found that the follow-up results and explanations addressed some of my concerns, and I also understand the resource limitations in running other retrieval-based decoding methods.
> >
> > I will increase my soundness score to 3.

---

> > > ### Author Response · Authors · 2024-08-13
> > >
> > > Dear Reviewer,
> > >
> > > We appreciate your recognition of our paper, and your suggestions have been valuable in enhancing it. We've shared you a new batch of results just in hope to inform you how our paper have progressed in last few days.
> > > Lastly, we sincerely thank you for investing your time in our work; your insights are invaluable to us. We will make sure that we incorporate your feedback in the next revision.  If you have any further unresolved questions, please feel free to point them out, and we will respond promptly.
> > >
> > > Please allow us to express our thankfulness once again, we sincerely appreciate your efforts.
> > >
> > > Best Regards,
> > >
> > > Paper 9447 Authors.

---

> ### Author Response · Authors · 2024-08-12
>
> **Dear Reviewer**,
>
> We sincerely appreciate your thorough review and valuable feedback. We have taken your comments seriously and have already conducted additional experiments and provided further explanatory details based on your recommendations.
>
> We've further measured our RTD methods on efficiency, both time and memory comsumption wise, here is our latest result.
>
> | Methods | Speed (it/s) | Extra Memory Usage (MB) |
> |-|-|-|
> | Baseline (PEFT) | $25.1$ | $0$ |
> | RTD | $23.6$ | ~$16$ |
> | ICL | $7.90$ | ~$37$ |
> | ICL + RTD | $7.85$ | ~$52$ |
>
> Considering both performance and efficiency analysis, **we can conclude that RTD offers a satisfactory solution for various task types due to its excellent cost-effectiveness, which makes it always worth trying**. A prefered workflow with RTD can be like:
> ``` verilog
> prompt tuning -> RTD -> ICL -> ICL+RTD -> finetune
> ```
>
> To ensure that we address the issues you raised effectively, we kindly request your prompt response. Your feedback is crucial for enhancing our work. We understand that the review process is demanding, and we greatly appreciate your time and effort.
>
> Thank you once a gain for your support and help.
>
> **Best regards,**
>
> **Authors**

---

### Author Response · Authors · 2024-08-14
**Conclusion and General Response of Paper 9447**

Dear PCs, SACs, ACs, and reviewers:

In our paper, we introduce the Reference Trustable Decoding (RTD) method and its multi-head variants as an efficient and low-cost approach to augmenting the capabilities of Large Language Models (LLMs) for downstream tasks. By leveraging an additional Datastore, we achieve the effect of editing the *LM_Head* layer without requiring additional training. Since our focus is on the later layers of the model, we gain greater flexibility and interpretability. Moreover, we avoid reliance on gradient-based backpropagation training methods, achieving rapid performance improvements or knowledge editing while maintaining efficiency. Our method incurs minimal additional cost during inference and benefits from mature optimization techniques. Compared to the excessively long KV Cache produced by ICL (In-Context Learning), our approach is more efficient. Additionally, Datastore generation is on par with inference costs, avoiding the need to retain optimizer parameters and intermediate states as in fine-tuning methods. Furthermore, it involves less computation and faster speed. We evaluate RTD across various benchmarks, including both language understanding and language generation tasks, as well as different multilingual models. RTD consistently performs on par with other augmentation methods.

We extend our heartfelt gratitude to all the contributors involved in reviewing this paper, including PCs, SACs, ACs, and all reviewers. Your time and effort are invaluable to us, and your insightful suggestions have significantly improved the quality of our work. In particular, we appreciate the feedback from reviewer *tphZ*, who raised questions about the applicability scope, prompting us to further evaluate the strengths and limitations of RTD, and the recognition of our work by the reviewer. We also acknowledge reviewer *wxwA* for highlighting issues related to generalization and trustable discussions in our original paper, which guided our paper’s optimization. Following discussions, the reviewer agreed to raise certain scores. Additionally, we thank reviewer *vZ3x* for their extensive inquiries; our timely responses facilitated valuable revisions to our article. Finally, if the reviewers could reconsider the scores by taking into account the additional content provided during the rebuttal and discussion stages, as well as our commitment to incorporating suggested improvements in the next revision, it would be a great honor for us.

Once again, we would like to express our gratitude to everyone who contributed to our paper. Your dedication and insights have been immeasurable for us.

Best Regards,

Paper 9447 Authors.

---

### Decision · Program_Chairs · 2024-09-25

**Decision:**

Accept (poster)

**Comment:**

This paper proposes Reference Trustable Decoding (RTD), a training-free paradigm for better adapting base LLMs to downstream tasks. RTD is a retrieval-based framework that creates a reference datastore from task datasets, storing key-value pairs of hidden states and corresponding labels. During inference, RTD retrieves the most relevant references based on the hidden states of the input and then aggregates them with model logits to compute the final output distribution. Experimental results demonstrate that RTD effectively adapts mainstream LLMs for natural language understanding (NLU) and natural language generation (NLG) benchmarks, outperforming ICL approaches.

The paper demonstrates good performance of RTD on multiple benchmarks, surpassing ICL methods and baselines. RTD is particularly meaningful in scenarios requiring rapid iteration of content or controllable generation. The methodology is clearly explained, and the figures provided help in visualizing the RTD process. The paper discusses the orthogonality of RTD with traditional methods, indicating its potential for combined usage and enhanced performance. The authors provide a thorough analysis of the hyper-parameters involved in RTD and their impact on performance.

Many of the weaknesses have been addressed by the authors in the rebuttal period and we would like to encourage the authors to include the results in the next revision of their paper. Related work regarding kNN-LM (e.g. [1]) should be more discussed in the paper. One of the last remaining questions is how RTD performs on out-of-distribution tasks.

[1] https://aclanthology.org/2023.acl-long.859/